# The Effect of Age on the Grouping of Open Clusters: II—Are There Old Binary Clusters?

Juan Casado

Facultad de Ciencias, Universidad Autónoma de Barcelona, 08193 Bellaterra, Barcelona, Spain; juan.casado@uab.cat or jcasadoo@hotmail.com

**Abstract:** In the present study, we continue testing the Primordial Group hypothesis (Casado 2022), which postulates that only sufficiently young open clusters can be binary or multiple, and old clusters are essentially single. To this end, we revisit all the remaining binary cluster candidates in the Galaxy having at least one cluster older than 100 Myr through Gaia data and careful revision of the literature. We found no convincing case for an old binary system among the 120 pairs/groups revised. Most of the pairs are optical pairs or flyby encounters. However, we found three dubious pairs that could falsify the title hypothesis upon further research. We also found two possible primordial pairs older than expected. Our results confirm that the vast majority of binary/multiple OCs in the Galaxy, if not all, are of primordial origin and are not stable for a long time. This finding is in line with similar studies of the Magellanic Clouds and theoretical N-body simulations in the Galaxy. The pairs of OCs in these groups are generally not binary systems since they are not gravitationally bound. We also point out some inconsistencies in previous works and databases, such as false open clusters and duplicities.

**Keywords:** open clusters and associations: general; galaxies: star formation; stars: color-magnitude (CMD) diagrams; stars: kinematics and dynamics; surveys; astrometry

## 1. Introduction

Open clusters (OCs) are born from the gravitational collapse of gas and dust within giant molecular clouds. There is observational evidence that at least some of them are born in groups [1–3], forming so-called primordial groups. Studies of grouping among OCs provide clues to understand star formation in the Galactic disc and the subsequent dynamical evolution of OCs. Moreover, they allow us to compare our Galaxy with nearby galaxies, such as the Magellanic Clouds, regarding these aspects.

OCs slowly disintegrate due to the tidal field of the Galaxy, interactions between their members, and close encounters with external stars and molecular clouds (e.g., Carraro [4]). Most clusters reach a maximum altitude above the Galactic plane during their orbits of less than 0.4 kpc, and only those older than 1 Gyr depart considerably from the Galactic plane, but typically less than 1 kpc [5]. These results suggest that older OCs may persist, among other reasons, because they undergo fewer interactions within the crowded thin disk during their Galactic orbits. After all, orbits outside the disk are one of the causes of the extraordinary longevity of globular clusters. On the other hand, the vertical distribution of young clusters is very flat [5]. Thus, young, low-interaction primordial groups appear to disperse across the Galactic disk in relatively short times (e.g., [6]).

Gravitational captures of OCs by other clusters are very rare and elusive events that could serve as laboratories to study the destruction of OCs [7,8]. Casado [3] listed 22 groups of double and multiple OC candidates in the Galactic sector from $l = 240°$ to $l = 270°$, involving 80 individual OCs. Unexpectedly, no plausible system with any member older than 0.1 Gyr was found. On this basis, the Primordial Group hypothesis postulates that only sufficiently young OCs can be multiple, and old OCs are essentially single,

since the gravitational interaction between OCs in primordial groups is very weak, and the probability of gravitational capture of two OCs without disruption or merger is very low [9]. In that first article of the present series, this postulate was defined and tested. First, we re-examined the work of de La Fuente Marcos and de La Fuente Marcos [6], where it was stated that only about 40% of the pairs of OCs are of primordial origin. However, we found no plausible binary system among their proposed OC pairs that had at least one member older than 0.1 Gyr. Some of the pairs are optical pairs, others are hyperbolic encounters, a few pairs appear to be primordial pairs with incorrect ages, and several of the putative OCs do not really exist. Second, we investigated the youngest OCs (age < 0.01 Gyr) in Tarricq et al. [5] and found that ~71% of them remain in their primordial groups. Third, a similar study of older OCs (age > 4 Gyr) shows that they are essentially alone. Fourth, the well-known case of the Perseus double cluster and some other candidates from the literature were also shown to fit the same hypothesis [9]. A simplified bimodal model allowed us to recover the global fraction of grouped OCs in the Galaxy (10–16%; [3,6]), assuming that young clusters remain associated ~0.04 Gyr on average [10–12]. This estimated fraction of clumped OCs is similar in the Large Magellanic Cloud [7]. These results suggest that most, if not all, of the groups are of primordial origin and are not stable for a long time, in line with similar conclusions obtained from the study of the Magellanic Clouds [7,13,14]. Hence, binary clusters encompassing old OCs should not exist.

However, NGC 1605 has recently been reported to consist of two clusters (one of intermediate age and one old) that merged via a flyby capture [15]. Nevertheless, both manual analysis and commonly used clustering analysis techniques show no indication of multiple populations in this cluster using the Gaia data [16]. Unexpectedly, the authors discovered a nearby OC, Can Batlló 1, with a similar age and distance but divergent proper motions (PMs), which may have been born in the same complex as NGC 1605.

The second Gaia Data Release (Gaia DR2) provided accurate astrometric data (positions, parallax, and PMs) and (1 + 2)-band photometry for about 1.3 billion stars [17], thus starting a new era in precision studies of Galactic OCs (among other subjects). The recent third release of Gaia early data results (Gaia EDR3) improves the precision of the measurements for around 1.8 billion sources [18]. The improved precision in parallax (*plx*), and particularly in PMs with respect to Gaia DR2 offers the opportunity to revisit the OC population and improve their characterization.

In the present study, we complete the test of the Primordial Group hypothesis by reviewing all remaining binary/multiple cluster candidates proposed in the literature and looking for counterexamples that may falsify this postulate. For this purpose, we used the Gaia EDR3 data and the numerous references throughout the existing literature. The study is outlined as follows. In Section 2, we describe the methodology used. In Section 3, we present the result set and discuss the remaining candidate pairs/groups that have member clusters > 0.1 Gyr; and in Section 4 we highlight some concluding remarks.

## 2. Methodology

The methods applied to manually select and study candidate OC pairs have been detailed previously [3,9]. However, we recapitulate here the general methodology.

We start with a candidate member of a hypothetical pair (or group) of Galactic OCs from the literature. For each of these candidates, we look for close correlations between coordinates, PMs and *plx* for all OCs within the studied area. For example, if two OCs are close enough (i.e., at a projected distance of less than 100 pc), the rest of their astrometric data are compared. If there is any overlap of the data, considering uncertainty intervals of 3σ, both OCs are included in Table 1. Table 1 is refined using the most accurate parameters of the individual OCs of the reports using Gaia data. When existing data are incomplete, or inconsistent between different authors, the OCs are manually re-examined using Gaia EDR3 to obtain the corresponding parameters. The Gaia EDR3 astrometric solution is accompanied by new quality indicators, such as the renormalized unit weight error (RUWE). RUWE allows sources with unreliable data to be discarded [18]. We systematically discarded

sources that have RUWE > 1.4. We also discarded sources with G > 18, unless otherwise stated, to limit the *plx* and PM errors, which increase exponentially with magnitude. The member stars of each re-examined OC are obtained using an iterative method that has been detailed previously [19]. In short, this method refines the approximate allowed ranges in position, PM and *plx*, initially obtained by eye, by examining the resulting CMD, which must include a maximum number of probable member stars but a negligible number of outliers (<3% of stars outside the OC's evolutionary sequences in its CMD). The error ranges in Table 1 are not standard uncertainties, but absolute errors that include all stars that are considered to be members of each OC at the end of the selection process. The main parameters of revisited OCs (age, distance, and extinction) are obtained using an artificial neural network (ANN) trained with objects of known parameters to compare with CMDs obtained from Gaia EDR3 photometry of the cluster members, following a method detailed elsewhere [20]. Thus, the age accuracy is ultimately tied to the reference values, which are derived from stellar evolution models by isochrone fitting. Isochrone fitting, the standard method to obtain such OC parameters, is often performed by hand and provides satisfactory results, but it is impractical to perform it on the samples of hundreds of OCs available from modern sky surveys. Using the ANN method, the uncertainty on the determination of log age ranges from 0.15 to 0.25 for young OCs and from 0.1 to 0.2 for old OCs. Following the criteria of previous studies, the obtained groups were refined by discarding OCs that are more than 100 pc away from any other member in 3D space [3,8,21,22]. This cutoff is an approximate and undemanding maximum, as other studies have used more restrictive limits (e.g., de La Fuente Marcos and de La Fuente Marcos [6] used 30 pc), but it is a convenient threshold for the most distant OCs detected by Gaia as their distance uncertainty reaches 0.1 kpc. Groups with radial velocity (RV) differences >10 km/s were also discarded [3,21]. Other authors have also used more restrictive limits (e.g., Soubiran et al. [8] used 5 km/s). Another requirement for refining the groups is $\Delta\text{PM}/plx$ (or $\Delta\text{PM}\,d$) < 2 yr$^{-1}$ between each pair of group members, using the units in Table 1. The latter condition implies that the differences in tangential velocities are also less than 10 km/s. The chosen limits are generous in order to obtain the most inclusive set of groups possible.

A straightforward way to search for possible OCs linked to each studied OC is to plot a graph of the Gaia sources that satisfy the examined OC constraints for the studied field. In this way, we can obtain plots similar to Figure 1, showing (or not) any related OCs. These plots are free of most of the noise from the unrelated field stars.

Data from the pre-Gaia literature are significantly less accurate than those from Gaia, especially when it comes to PMs (excluded from the present analysis) and ages [23]. Nevertheless, many of the reported data on *d*, RV and even the age of well-studied objects have some value. Therefore, they are sometimes used in the individual discussion of candidate pairs for comparison with Gaia data.

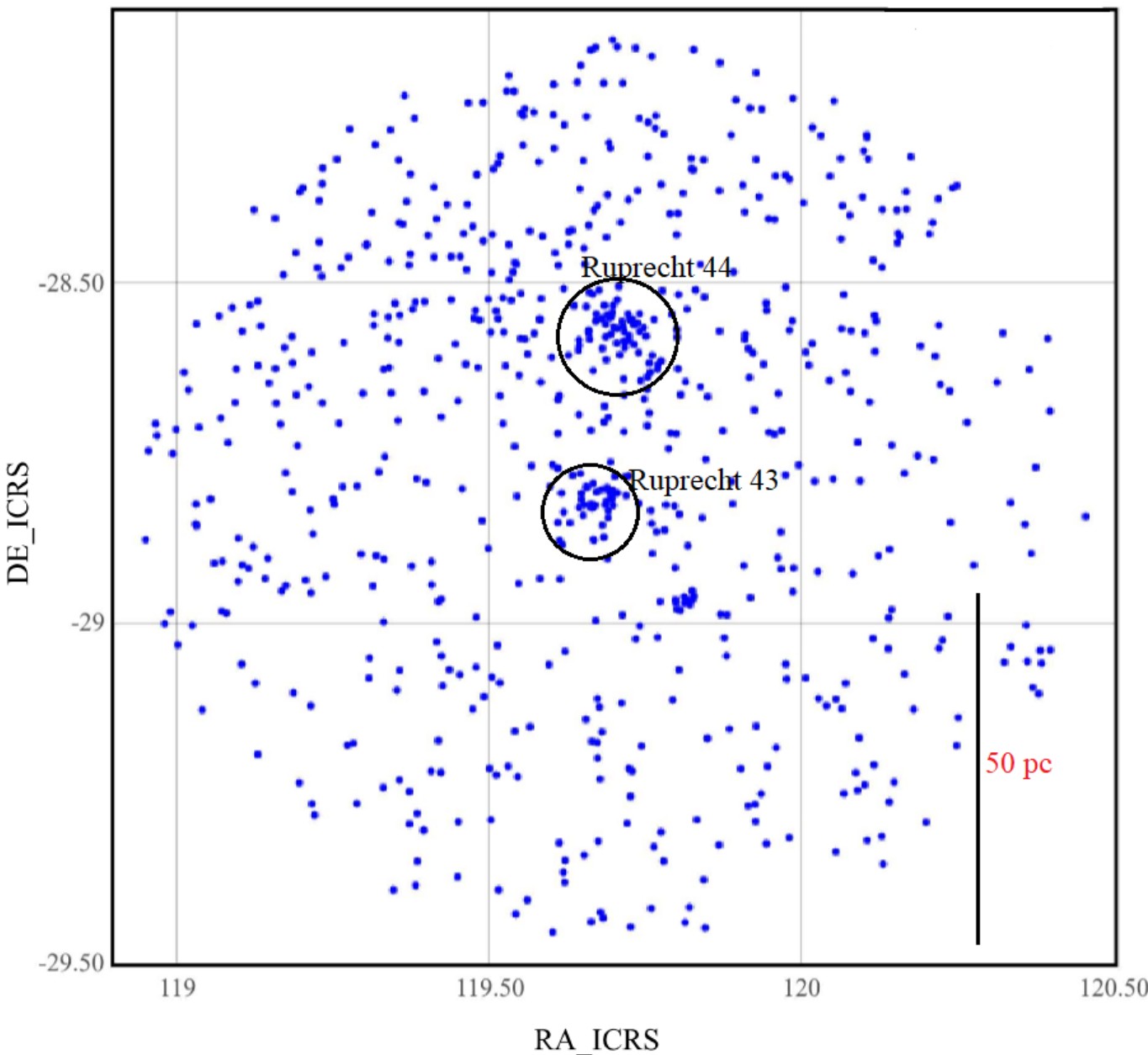

**Figure 1.** Chart of selected Gaia EDR3 sources defining the candidate group Y. Constraints: *plx* 0.10 to 0.215 mas; $\mu_\alpha$ −2.0 to −2.4 mas yr$^{-1}$; $\mu_\delta$ 2.9 to 3.3 mas yr$^{-1}$; RUWE < 1.4; *G* < 19. The projected distance bar assumes a heliocentric distance to the group of 5.5 kpc (see text).

**Table 1.** Selected possible OC groups (Gr) and candidate member's properties studied in this work. Column headings: 1. Group label; 2. OC name; 3. Galactic longitude; 4. Galactic latitude; 5. Parallax; 6. Photometric Distance; 7. PM in right ascension; 8. PM in declination; 9. OC radius; 10. Number of member stars; 11. Age; 12. Radial velocity. Abbreviations: [a] radius containing 50% of members; [c] maximum cluster member's distance to average position; [e] reexamined using Gaia EDR3 due to insufficient, imprecise or inconsistent reports; [f] too few stars for a complete characterization; [g] see text; [p] protocluster or embedded cluster; ? dubious.

| 1 | 2 | 3 | 4 | 5 | 6 | 7 | 8 | 9 | 10 | 11 | 12 | 13 |
|---|---|---|---|---|---|---|---|---|---|---|---|---|
| Gr | OC | $l$ | $b$ | $plx$ | $d$ | $\mu_\alpha$ | $\mu_\delta$ | $R$ | $N$ | Age | RV | References and notes |
| | Name | degree | degree | mas | kpc | mas/yr | mas/yr | arcmin | stars | Gyr | km/s | |
| P ? | Haffner 10 | 230.80 | 1.02 | $0.26 \pm 0.06$ | 3.3–3.7 [g] | $-1.1_5 \pm 0.2$ | $1.6 \pm 0.1$ | 2.5 | 82 | 1.1–5 [g] | $88._2 \pm 6$ | This work [e] |
| P ? | Czernik 29 | 230.81 | 0.95 | $0.24 \pm 0.06$ | 2.9–3.5 [g] | $-1.6 \pm 0.1$ | $1.3 \pm 0.2$ | 2.5 | 35 | 0.08–0.17 [g] | $88._4 \pm 6$ | This work [e] |
| - | NGC 1907 | 172.62 | 0.32 | $0.61_1$ | $1.6_2$ $1.5_7$ | $-0.0_4$ | $-3.4$ | 4.5 [a] | 275 | 0.59 0.56 | - $3._1$ | [20] [5] |
| - | NGC 1912 | 172.27 | 0.68 | $0.87_4$ | $1.1_0$ $1.0_8$ | 1.6 | $-4.4$ | 9.8 [a] | 349 | 0.29 0.30 | - $-0._3$ | [20] [5] |
| Q ? | ASCC 14 | 171.83 | $-1.03$ | $0.93 \pm 0.07$ | 1.1 [g] | $-0.4_5 \pm 0.2_3$ | $-3.4_1 \pm 0.4_6$ | 55 13 | 57 | $0.06_4$–0.41 [g] | - $-17$ [g] | This work [e] [24] |
| Q ? | Gulliver 8 | 173.21 | $-1.55$ | $0.87_2$ $0.90 \pm 0.11$ | $1.1_1$– $1.1_7$ [g] | $-0.2$ $-0.3 \pm 0.2$ | $-3.0$ $-3.1 \pm 0.2$ | 6.1 [a] 14 | 29 32 | $0.02_1$–$0.06_3$ [g] | $-2$– 34 [g] | [20] This work [e] |
| - | NGC 6281 | 347.76 | 1.99 | 1.89 | $0.53_9$ $0.53_2$ | $-1.8_9$ | $-4.0_4$ | 16 [a] | 488 | 0.51 0.51 | - $-5$ | [25] [5] |
| - | NGC 6405 | 356.58 | $-0.76$ | 2.18 | $0.45_9$ $0.45_4$ | $-1.3_5$ | $-5.8_4$ | 17 [a] | 567 | $0.03_4$ $0.03_6$ | - $-7$ | [25] [5] |
| - | Turner 9 | 64.94 | 2.64 | $0.57_2$ | $1.7_7$ $1.7_0$ | $0.3_4$ | $-2.4_3$ | 9.1 [a] | 34 | $0.30_9$ $0.32_4$ | - $-19$ | [25] [5] |
| - | ASCC 110 | 70.41 | 1.38 | $0.50_7$ | $1.9_7$ $1.8_9$ | $0.3_0$ | $-3.1_8$ | 12 [a] | 65 | 0.79 0.87 | - 10 | [25] [5] |
| - | NGC 7438 (=UBC 175) | 106.78 | $-4.87$ | $0.73_9$ | $1.3_3$ $1.3_0$ | $1.4_5$ | $-0.8_0$ | 9.2 [a] | 58 | 0.47 0.51 | - $-28$ | [25] [5] |
| - | ASCC 115 | 97.53 | $-2.50$ | 1.31 | $0.73_1$ | $-0.5_6$ | $-0.6_4$ | 15 [a] | 32 | $0.13_2$ | $-9$– $-46$ [g] | [25] |
| - | NGC 4349 | 299.73 | 0.84 | $0.49_3$ | 1.7–2.2 [g] | $-7.8$ | $-0.2$ | 17 [c] | 749 | 0.98 | $-12$– $-16$ [g] | [22] |
| - | Patchick 57 (=UBC 284) | 299.03 | $-0.35$ | 0.50 | 1.8 | $-7.1$ | $-1.2$ | 12 [c] | 222 | $2.0_4$ $1.7_4$ | $-39$– $-63$ [g] | [22] [5] |

**Table 1.** *Cont.*

| 1 | 2 | 3 | 4 | 5 | 6 | 7 | 8 | 9 | 10 | 11 | 12 | 13 |
|---|---|---|---|---|---|---|---|---|---|---|---|---|
| Gr | OC | $l$ | $b$ | $plx$ | $d$ | $\mu_\alpha$ | $\mu_\delta$ | $R$ | $N$ | Age | RV | References and notes |
| | Name | degree | degree | mas | kpc | mas/yr | mas/yr | arcmin | stars | Gyr | km/s | |
| R ? | NGC 129 | 120.32 | −2.55 | $0.50_7$ $0.52_8$ | 1.8–1.9 [g] | $-2.5_9$ $-2.5_9$ | $-1.1_2$ $-1.1_8$ | 25 [c] 10 [a] | 368 363 | $0.98$–$0.07_2$ [g] | −38– −40 [g] | [22] [25] |
| R ? | Stock 21 | 120.14 120.13 | −4.82 −4.84 | $0.50_2$ $0.47_9$ | 1.7– $2.0_4$ [g] | $-2.1_5$ $-2.1_4$ | $-1.7_8$ $-1.8_9$ | 20 [c] 3.1 [a] | 93 57 | 0.48–0.28 [g] | −35– −63 [g] | [22] [25] |
| - | Czernik 2 | 121.98 | −2.68 | $0.49_6$ | 1.7–1.9 [g] | $-4.0_5$ | $-0.9_6$ | 10 [c] | 95 | $1.5_9$ | - | [22] |
| - | LP 1970 | 183.44 183.45 | 3.49 3.46 | $0.24_8$ $0.23_6$ | $3.4_3$ | $0.2_0$ $0.2_0$ | $-1.5_2$ $-1.5_3$ | 17 [c] 11 [a] | 90 416 | $1.4_1$ 0.68 | 53– 30 [g] | [22] [26] |
| - | LP 1377 | 182.35 182.34 182.31 | 3.10 3.10 2.95 | $0.24_6$ $0.27_5$ $0.24 \pm 0.08$ | - - - | $0.4_4$ $0.4_1$ $0.3 \pm 0.3$ | $-1.4_3$ $-1.3_9$ $-1.7 \pm 0.2$ | 20 [c] - 1.4 | 117 117 12 [f] | 0.98 $0.09_8$ | - −22 - | [22] [27] This work [e] |
| S ? | NGC 2354 | 238.40 238.38 | −6.85 −6.84 | 0.75 $0.75_3$ | $1.3_7$ $1.3_3$ | −2.9 −2.89 | 1.8 1.83 | 42 [c] 9.1 [a] | 238 265 | 1.18 1.41 1.45 | 34 | [22] [20] [5] |
| S ? | NGC 2362 | 238.16 238.18 | −5.54 −5.55 | $0.74_8$ $0.74_2$ | $1.3_4$ $1.3_0$ | −2.77 −2.79 | 2.95 2.95 | 22 [c] 3.1 [a] | 194 144 | $0.00_8$ $0.00_6$ $0.00_6$ | 29 | [22] [20] [5] |
| T ? | NGC 581 | 128.05 | −1.80 | $0.37_3$ | $2.5_0$ | −1.38 | −0.50 | 3.7 [a] | 131 | $0.02_8$ | −40 – −46 [g] | [20] |
| T ? | NGC 663 | 129.49 | −0.96 | $0.32_0$ | $2.9_5$ | −1.11 | −0.23 | 9.0 [a] | 468 | 0.03– 1.1 [g] | −32– −44 [g] | [20] |
| - | COIN-Gaia 4 | 129.78 | −3.41 | 0.45 | $2.2_1$ $2.1_1$ | −0.95 | −1.00 | 6.4 [a] | 42 | $0.05_4$ $0.06_6$ | −5 | [20] [5] |
| - | NGC 659 | 129.38 | −1.53 | $0.28_7$ | $3.3_2$ $3.1_4$ | −0.80 | −0.26 | 3.8 [a] | 138 | $0.01_5$ $0.01_7$ | 78– −65 [g] | [20] [5] |
| - | Ruprecht 100 | 297.75 297.75 | −0.21 −0.22 | $0.27_4$ $0.29 \pm 0.06$ | $3.3_7$– $2.7_8$ [g] | $-8.2_7$ $-8.3 \pm 0.3$ | $0.9_3$ $0.8 \pm 0.3$ | 2.8 [a] 3.2 | 47 74 | $0.20_4$ | −7– −11 | [25] This work [e] |
| - | Ruprecht 101 | 298.20 | −0.51 | $0.28_6$ | $2.7_1$– $2.8_5$ [g] | $-9.5_1$ | $0.4_8$ | 5.0 [a] | 255 | $1.0_5$ | 11– −63 [g] | [25] |

**Table 1.** *Cont.*

| 1 | 2 | 3 | 4 | 5 | 6 | 7 | 8 | 9 | 10 | 11 | 12 | 13 |
|---|---|---|---|---|---|---|---|---|---|---|---|---|
| Gr | OC | $l$ | $b$ | $plx$ | $d$ | $\mu_\alpha$ | $\mu_\delta$ | $R$ | $N$ | Age | RV | References and notes |
| | Name | degree | degree | mas | kpc | mas/yr | mas/yr | arcmin | stars | Gyr | km/s | |
| U ? | Theia 847 | 152.33 | 6.37 | 1.62<br>$1.62 \pm 0.16$ | $0.65_7$<br>0.64 | $-0.9_2$<br>$-1.0_8 \pm 0.3$ | $-1.0_9$<br>$-1.0_5 \pm 0.5_5$ | 33 [a]<br>37 | 158<br>110 | 0.35<br>0.26 | -<br>$-10 \pm 4$ | [25]<br>This work [e] |
| U ? | UPK 312 | 150.32<br>150.35 | 5.42<br>5.44 | $1.42_5$<br>$1.42 \pm 0.11$ | 0.72<br>$0.68_4$ | $0.1_3$<br>$0.0 \pm 0.6$ | $-0.7_6$<br>$-0.8_5 \pm 0.6_5$ | $15._6$ [a]<br>25 | 29<br>31 | 0.47<br>0.33 | -<br>$-2 \pm 7$ [g] | [25]<br>This work [e] |
| - | LP 2117 | 28.70<br>28.71 | $-1.54$<br>$-1.55$ | 0.61<br>$0.63 \pm 0.13$ | $1.5_1$ | $1.0_9$<br>$1.0_3 \pm 0.3_5$ | $0.0_9$<br>$0.0_7 \pm 0.3_2$ | 11 [a]<br>14 | 526<br>518 | 0.63 | $-8-$<br>$-10$ [g] | [20]<br>This work [e] |
| - | UBC 354 | 28.78 | $-1.56$ | 0.54<br>0.61 | $1.8_6$ | $1.2_5$<br>$1.1_9$ | $0.1_3$<br>$0.0_8$ | 5.7 [a] | 22<br>21 | 0.20 | $-9-$<br>$-10$ [g] | [20]<br>[25] |
| - | ASCC 88 | 350.01 | 2.97 | 1.11 | $0.9_3$ [g] | $0.9_0$ | $-3.2_8$ | 22 [a] | 119 | 0.25 | $-38-3$ [g] | [25] |
| - | Gulliver 29 | 350.23 | 3.28 | 0.91<br>$0.90 \pm 0.09$ | $1.1_8$ [g] | $1.2_9$<br>$1.2 \pm 0.4$ | $-2.2_1$<br>$-2.2 \pm 0.5$ | 40 [a]<br>39 | 341<br>315 | $0.01_1$ | $1-29$ [g] | [25]<br>This work [e] |
| - | Barkhatova 1 | 86.21 | 0.82 | $0.48_9$ | $2.2-1.9$ [g] | $-2.4_5$ | $-4.4_8$ | 4.3 [a] | 47 | 0.26 | $-37-$<br>$-1$ [g] | [25] |
| - | Gulliver 30 | 86.30 | 0.65 | $0.44_2$ | $2.3-2.0$ [g] | $-2.5_3$ | $-3.7_0$ | 5.0 [a] | 49 | 0.15 | $-10-14$ [g] | [25] |
| - | NGC 2168 (=M35) | 186.61 | 2.23 | 1.15 | 0.91<br>0.89 | $2.2_6$ | $-2.8_9$ | 19 [a] | 1239 | $0.06-0.40$ [g] | -<br>$-8$ | [25]<br>[5] |
| - | Coin-Gaia 24 | 186.89 | 0.42 | $0.98_8$ | 1.03 | $2.5_2$ | $-2.9_6$ | 12 [a] | 47 | $0.06-0.25$ [g] | -<br>23 [g] | [25] |
| V | Kronberger 1 | 173.11 | 0.05 | $0.44_3$ | $2.3_4$ | $-0.0_5$ | $-2.2_0$ | 0.8 [a] | 32 | $0.00_6$ | $-21-13$ [g] | [20] |
| - | Gulliver 53 | 173.23<br>173.24 | $-1.14$<br>$-1.14$ | $0.38_4$<br>$0.40 \pm 0.13$ | $2.4_3$ | $0.4_0$<br>$0.4 \pm 0.3$ | $-2.8_4$<br>$-2.8 \pm 0.6$ | 7.4 [a]<br>10 | 23<br>87 | $0.12_8$ | $-36-30$ [g] | [20]<br>This work [e] |
| V | Stock 8 | 173.32 | $-0.22$ | $0.44_6$ | $2.3_5$ | $0.0_9$ | $-2.2_5$ | 9.2 [a] | 275 | $0.01_4$ | $-18-40$ [g] | [20] |
| V | NGC 1931 | 173.90 | 0.27 | $0.41 \pm 0.12$ | - | $0.1 \pm 0.4$ | $-2.0 \pm 0.4$ | 4.5 | 65 | $0.00_2-0.01_2$ [g] | $-23$ | This work [p,e] |
| - | Coin-Gaia 40 | 174.05 | $-0.79$ | 0.47 | $1.9_2$ [g] | $0.3_9$ | $-2.7_6$ | 4.4 [a] | 12 [f] | 0.14 [g] | | [20] |
| - | Collinder 220 | 284.55 | $-0.35$ | $0.38_6$ | $2.4_6$<br>$2.3_6$ | $-7.3_7$ | $2.6_8$ | 6.5 [a] | 110 | 0.23<br>0.28 | -<br>$-13$ | [25]<br>[5] |

**Table 1.** *Cont.*

| 1 | 2 | 3 | 4 | 5 | 6 | 7 | 8 | 9 | 10 | 11 | 12 | 13 |
|---|---|---|---|---|---|---|---|---|---|---|---|---|
| Gr | OC | $l$ | $b$ | *plx* | $d$ | $\mu_\alpha$ | $\mu_\delta$ | $R$ | $N$ | *Age* | RV | References and notes |
| | Name | degree | degree | mas | kpc | mas/yr | mas/yr | arcmin | stars | Gyr | km/s | |
| - | IC 2581 | 284.59 | 0.03 | $0.37_3$ | $2.5_4$ $2.4_1$ | $-7.2_8$ | $3.6_0$ | 3.5 [a] | 101 | $0.01_0$ $0.01_2$ | - 67 | [25] [5] |
| - | FSR 448 | 115.21 | −1.00 | $0.32_2$ | $3.2_2$ $2.9_3$ | $-2.4_6$ | $-2.0_9$ | 3.2 [a] | 32 | 0.39 0.46 | - −51 | [25] [5] |
| - | FSR 451 | 115.75 115.69 | −1.12 −1.15 | $0.33_1$ $0.35 \pm 0.09$ | $3.2_5$ $3.1_0$ | $-3.2_1$ - $-3.2 \pm 0.4$ | $-1.9_6$ - $-2.0 \pm 0.3$ | $12._8$ [a] - 11 | 160 - 227 | $0.01_3$ $0.01_4$ | - −81 −76 | [25] [5] This work [e] |
| W ? | NGC 2345 | 226.59 | −2.32 | $0.35_6$ | $2.6_6$ $2.5_5$ | $-1.3_5$ | $1.3_7$ | 5.4 [a] | 409 | $0.20_9$ $0.21_4$ | 54–63 [g] | [25] [5] |
| W ? | FSR 1207 | 226.61 | −2.54 | $0.38_0$ $0.40 \pm 0.10$ | $2.5_1$ $2.5_8$ | $-1.3_3$ $-1.3 \pm 0.2$ | $0.7_8$ $0.7_6 \pm 0.2$ | 1.4 [a] 4.5 | 43 37 | 0.24 $0.15_1$ | 29 [g] - | [25] This work [e] |
| - | Pismis 19 | 314.71 | −0.31 | 0.26 | 3.5 | $-5.5$ | $-3.2$ | 2.1 [a] | 430 | $0.63-1.1_2$ [g] | - $-29._6$ | [20] [5] |
| X | Trumpler 22 | 314.66 | −0.59 | 0.39 0.41 | 2.4 2.4 $2.3_6$ | $-5.1$ $-5.1_5 \pm 0.2$ | $-2.7$ $-2.7 \pm 0.2$ | 6.2 [a] 8.8 | 140 183 | $0.02_4-0.31$ [g] 0.13 | −38– −43 [g] −44 ± 2 | [20] [5] This work [e] |
| X ? | NGC 5617 | 314.68 | −0.11 | 0.40 | $2.2_4$ $2.1_6$ | $-5.6_4$ | $-3.1_7$ | 7.0 [a] | 565 | $0.10_4$ $0.10_9$ | −39– −35 [g] | [20] [5] |
| X | Hogg 17 | 314.90 | −0.86 | $0.40_7$ | $2.2_8$ $2.1_6$ | $-5.2_0$ | $-2.4_3$ | 3.5 [a] | 41 | 0.07–0.18 [g] | - −44 | [20] [5] |
| - | NGC 2194 | 197.25 | −2.34 | $0.28_2$ | $3.4_9$ $3.3_2$ | $0.4_9$ | $-1.4_3$ | 4.6 [a] | 735 | $0.33_9$ $0.34_7$ | - 39 | [25] [5] |
| - | Skiff J0614 + 12.9 | 197.31 | −2.10 | $0.27_0$ | $3.5_0$ $3.2_4$ | $0.9_2$ | $-1.9_9$ | 1.6 [a] | 46 | $0.30_2$ $0.37_2$ | - 46 | [25] [5] |
| Y | Ruprecht 44 | 245.73 | 0.49 | $0.19 \pm 0.04$ | 4.6–5.8 [g] | $-2.4 \pm 0.2$ | $2.9 \pm 0.2$ | 5.0 | 95 | 0.01 [g] | 71–94 [g] | [3] |
| Y | Ruprecht 43 | 245.93 | 0.36 | $0.16 \pm 0.06$ | 5.5– $0.9_7$ [g] | $-2.2 \pm 0.2$ | $3.1 \pm 0.2$ | 4.0 | 36 | $0.05_7-0.25$ [g] | 94–113 [g] | This work [e] |

### 3. Analysis of Candidate OC Pairs/Groups from the Literature Having at Least One Member > 0.1 Gyr

*3.1. Candidate Pairs/Groups Suggested before Gaia*

3.1.1. Haffner 10/Czernik 29

This pair was first proposed as a candidate binary cluster by FitzGerald and Moffat [28]. The putative components have very different ages: Gaia-derived ages for Haffner 10 range between 1.1 Gyr [22] and 5 Gyr [26], while Gaia studies on Czernik 29 report values between 0.08 Gyr [29] and 0.17 Gyr [5]. Therefore, they cannot be a primordial pair due to their age difference. The review of their data in Gaia EDR3 (Table 1) shows a dubious picture. The mean parallaxes are comparable. However, the reported parallaxes of 0.235 mas for Czernik 29 [20] and 0.274 mas for Haffner 10 [30] suggest that Haffner 10 would be closer to the Sun. On the other hand, the photometric *d* shows the opposite trend: For Haffner 10, most estimates range between 3.3 kpc [5] and 3.7 kpc [24,26], while for Czernik 29, *d* range from 2.9 kpc [28] to 3.5 kpc [20,31]. The PMs imply that the tangential velocity between them is slightly higher than the adopted threshold of 10 km/s, assuming an average parallax of 0.25 mas. The RVs just released from Gaia DR3 help in this case, as they are well-matched: $88.4 \pm 6$ km/s for 21 members of Czernik 29 and $88.2 \pm 6$ for 23 members of Haffner 10. The orbital elements derived from 6D-astrometry [5] are not precise enough to settle the question, although the orbits of both OCs around the Galactic center are different. Last but not least, plots similar to Figure 1 do not show any relation between them. All in all, the current evidence is not enough to either confirm or rule out this uncertain pair.

3.1.2. NGC 1907/NGC 1912

This intermediate-age pair of OCs was proposed by Subramaniam et al. [32] and studied in detail by Subramaniam and Sagar [33]. They concluded that both OCs have similar distances and ages, suggesting that they may have been born together, thus reinforcing the candidacy for a binary system. However, they found that the tidal timescale for this pair is shorter than their age, suggesting that the clusters are not gravitationally bound. De Oliveira et al. [34] used *N*-body simulations to study the dynamical state of this pair of clusters and theorized the formation of stellar bridges between them. The authors suggested that the clusters were born in different regions of the Galaxy and are currently undergoing a flyby. Although both OCs were included in Conrad et al. [21], this study does not report their supposed association.

In contrast, the high-quality Gaia EDR3 data allow us to reliably conclude that the two clusters do not form a binary system. In the unlikely event that they were born together, they separated from each other long ago (Table 1). All relevant data are unrelated, with the possible exception of the RVs. Both *plx* and photometric *d* indicate that NGC 1907 is about 500 pc further away than NGC 1912, as suggested by the mere inspection of their photographic images, and there are no physical stellar bridges between them. Their age difference (Table 1) is therefore not surprising.

3.1.3. NGC 1912/ASCC 14/Gulliver 8

Instead of NGC 1907 (see above), Conrad et al. [21] suggested that NGC 1912 might be associated with ASCC 14 (or (KPR2005) 14). The only Gaia-based report on ASCC 14 is by Hao et al. [27]. However, the parameters listed in that catalog are inconsistent with previous literature. For example, *plx* (0.41 mas) is at odds with d (1.1 kpc; e.g., [24,35]). The only retrieved ages of ASCC 14 were 0.34 Gyr [35] and 0.41 Gyr [36], and the only RV recovered was $-16.9 \pm 10$ km/s (e.g., [24]), from a single star. Therefore, we revisited this object using Gaia EDR3 and confirmed a dispersed and perhaps dissolving OC, whose main parameters are summarized in Table 1. The mean parallax (0.92 mas; parallaxes in Table 1 only reflect the range of members parallaxes) agrees with the reported d, but the PMs are in disagreement (as usual) with the pre-Gaia reports on ASCC 14. Nevertheless, the resulting CMD shows a well-defined main sequence (Figure 2). The inconsistency of

the PMs and RVs of NGC 1912 and ASCC 14 is observed in Table 1 and allows us to rule out with high probability that they form a physical system.

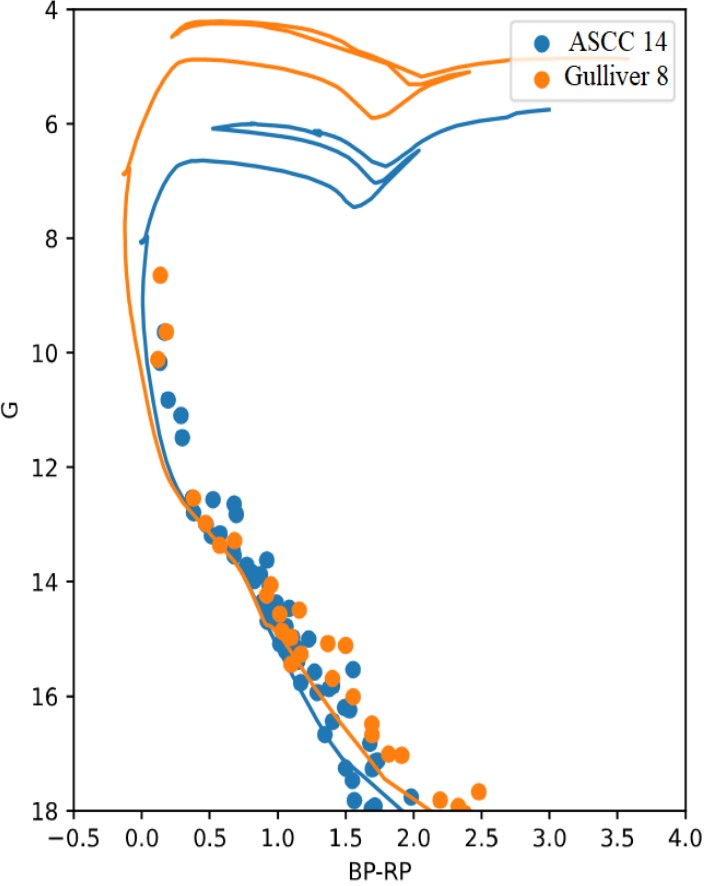

**Figure 2.** Overlapping CMDs and fitting isochrones of ASCC 14 and Gulliver 8, obtained using Gaia EDR3 photometry. The constraints are defined in Table 1.

On the other hand, our study of the stellar field around ASCC 14 suggested that Gulliver 8 could be associated with it. As shown in Table 1, all Gaia astrometric data seem to be compatible for both OCs. The reported photometric distances of Gulliver 8 are 1.11 kpc [26] and 1.14 kpc [20]. All *plx*, including our median *plx* (0.92 mas), agree with a distance of $1.11 \pm 0.03$ kpc. Thus, the heliocentric distances of the two OCs are well-matched. The projected distance between them is ~28 pc, an acceptable value for an interacting system. However, the two retrieved RVs: $-1.6 \pm 2.9$ km/s [37] and 33.8 km/s [38] do not match the reported RV of ASCC 14. The question of age is also relevant, since Gulliver 8 is a young OC with reported ages from 21 Myr [20] to 63 Myr [27]. Our estimation is ~42 Myr. On the other hand, ASCC 14 appears to be, in principle, an intermediate-age OC (see, however, the comment on the ages in Kharchenko's catalog in Group X discussion). Nevertheless, our determination of the age of ASCC 14 (~64 Myr) and the fact that the CMDs of both OCs are alike (Figure 2) suggest that they may form a primordial pair of relatively young clusters (Group Q). This possibility, however, needs to be confirmed by new RV measurements. Unfortunately, Gaia DR3 does not show enough definitive RVs in this case.

### 3.1.4. NGC 1528/NGC 1545

This pair was considered as a probable binary system by Loktin [39] as its relative antiquity suggested a high probability of gravitational binding in this system. However, analysis of it using Gaia reveals that this statement is incorrect as the PM, *plx*, and *d* of each OC are in disagreement [20], even if their RVs are compatible [5]. The Gaia results

indicate that NGC 1528 is more than 300 pc further away than NGC 1545, so it is merely an optical pair. Consequently, the CMDs do not match [20] and the ages are also different: Gaia-derived ages converge in the interval 0.10 to 0.12 Gyr for NGC 1545 [5,20,40,41], but range from 0.29 to 0.39 Gyr for NGC 1528 [5,20,22,26,40,41].

### 3.1.5. Loden 915/ESO 175-6

This pair was proposed by Conrad et al. [21]. Gaia data of Loden 915 have been listed only by Hao et al. [27], and its reported *plx* (0.365 mas) is strikingly at odds with the two reported photometric *d*: 500 pc [24] and 709 pc [35]. We have not found evidence of this OC in Gaia EDR3 data. Moreover, pre-Gaia data are dispersed and scarce. For instance, only the Kharchenko group provided an age value, and only one RV has been reported [21]. Therefore, we believe this is only a mirage instead of a real OC.

Anyway, even if Loden 915 exists and the data used by Conrad et al. [21] are correct, we obtain a relative velocity >10 km/s, mainly due to the disparate $\mu_\delta$ of both OCs: 1.00 mas/yr for Loden 915 and −2.43 mas/yr for ESO 175-6. All things considered, it is safe to say that this is not a physical pair of OCs, at least according to our criteria.

### 3.1.6. NGC 6281/NGC 6405

This pair was proposed by Conrad et al. [21]. However, Gaia data significantly modified the mean parameters of these OCs. Although RVs are close enough, PMs are only marginally compatible (Table 1). The distance of NGC 6281 is well known: All Gaia-derived photometric *d* agree on 531 ± 8 pc. The corresponding parallaxes converge on a concurrent distance of 529 ± 6 pc. For NGC 6405, we have a consistent *d* of 459 ± 6 pc, and all parallaxes agree on 457 ± 3 pc. Thus, the heliocentric distance difference between the two OCs is ~70 pc. Since the projected distance between them is ~80 pc, the two OCs are >100 pc apart. Therefore, they cannot form an interacting pair according to our criteria.

A possible relation between NGC 6405 and ASCC 90 has been discarded previously [9].

### 3.1.7. Turner 9/ASCC 110

Conrad et al. [21] found this candidate pair. Although RVs in Table 1 are at odds, this is not enough to ascertain the nature of this pair, as reported RVs of both OCs are miscellaneous (e.g., Conrad et al. [21]). However, the Gaia-derived distances are helpful in this case. Photometric *d* of Turner 9 ranges from 1.61 kpc [26] to 1.77 kpc [20], and all parallaxes, except the measurement in Hao et al. [27], including our median *plx* (0.57 mas) point to a consistent distance of 1.72 ± 0.03 kpc. Regarding ASCC 110, *d* range from 1.58 kpc [26] to 1.97 kpc [20], and six out of eight *plx* indicate a compatible distance of 1.93 ± 0.04 kpc. Thus, although photometric estimations are partially overlapping for the two OCs, Gaia mean parallaxes clearly show that both OCs are at least 140 pc apart and, therefore, they form merely an optical pair.

### 3.1.8. NGC 7438/ASCC 115

Conrad et al. [21] proposed this candidate pair. However, the data used by these authors only five years ago were outdated by Gaia data (Table 1): Almost all the relevant parameters of the two OCs, except $\mu_\delta$, are disparate. RVs could be compatible, but they are too assorted for ASCC 15, ranging from −9.0 km/s (e.g., [21,24]) to −46.0 km/s [8]. Therefore, this is a mistaken candidate pair. Note that NGC 7438 has been rediscovered as UBC 175 [20].

### 3.2. Candidate Pairs/Groups Suggested by Liu and Pang

Liu and Pang [22] searched for OC groups in their catalog using the Friend-of-Friend algorithm. The criterion used was a linking length ≤100 pc based solely on the 3D positions of OCs. They listed 52 candidate groups comprising a total of 152 OCs as members. We have reviewed their classification for groups that have at least one member older than 0.1 Gyr and most of these groups have been discarded as their PMs are incompatible, even

considering experimental uncertainty at a $3\sigma$ level, suggesting that they would be flyby events. However, there are some cases that deserve a more detailed analysis:

### 3.2.1. NGC 4349/Patchick 57

These OCs have similar astrometric parameters at first sight (Table 1). The range of reported *d* of NGC 4349 is relatively consistent: from 1.7 kpc [26] to 2.2 kpc [24,42]. All parallaxes point to a coherent distance between 1.90 kpc [25] and 1.93 kpc [20,26]. Patchick 57 was listed as an OC remnant candidate by Bica et al. [43] and later identified as a new OC: UBC 284 [20]. Its only retrieved *d* are 1.78 kpc [5] and 1.87 kpc [20], while all reported parallaxes agree on a concurrent distance of 1.88 kpc. Therefore, both OCs could be at a coincident distance. However, there is a significant difference between their PMs (1.22 mas/y), which combined with the average parallax of the pair leads to a relative tangential velocity of 12 km/s, higher than the adopted limit of 10 km/s. Furthermore, Patchick 57 has reported RVs of $-39.4$ km/s [5,27] and $-62.9$ km/s [38], which are at odds with the RVs of NGC 4349: from $-11.5$ km/s [8,27] to $-15.7$ km/s (e.g., [42]). Ultimately, this duo should be considered a chance alignment or a hyperbolic encounter.

### 3.2.2. NGC 129/Stock 21/Czernik 2

Most of the astrometric parameters of the first pair are well fitted and the PMs are marginally compatible (Table 1). The Gaia photometric distances are also compatible: For NGC 129, all *d* are between 1.80 kpc [26] and 1.91 kpc [20], while for Stock 21 these distances range from 1.71 kpc [26] to 2.04 kpc [20]. The corrected parallaxes of NGC 129 converge to a distance of $1.87 \pm 0.02$ kpc. However, something odd happens with the parallaxes of Stock 21: the corrected Gaia DR2 values point to a distance compatible with NGC 129 ($1.85 \pm 0.04$ kpc), but Gaia EDR3 data suggest a larger distance (2.1 kpc), in which case Stock 21 would be more than 100 pc away from NGC 129. Moreover, from the positions and parallaxes in Table 1, the projected distance between the two OCs would be ~80 pc, high but insufficient to rule out this pair.

Similarly, the PMs allow us to estimate an acceptable tangential velocity of ~8 km/s. More doubts arise when we examine the RVs in the literature. While the RV of NGC 129 appear to be between $-37.9$ km/s [8] and $-40.1$ km/s [5], Stock 21 has varied records: from $-34.6$ km/s [5] to $-62.8$ km/s [37]. If the former value is correct, we would have a dubious candidate, but if the latter value is correct, this pair should be discarded. Gaia DR3 provides a RV of $-35.7$ km/s for only one member star.

The age of NGC 129 is ill-defined: Gaia-derived ages range from 0.98 Gyr [22] to 0.072 Gyr [26]. Such a wide range encompasses the reliable Stock 21 ages: from 0.48 Gyr [22] to 0.28 Gyr [20]. Therefore, both OCs could be of similar age and this could be a rare case of a primordial pair of intermediate age. All these scenarios are open so far but, all things considered, this pair seems, to say the least, dubious.

As for Czernik 2, the Gaia *d* range from 1.69 kpc [26] to 1.90 kpc [20], the latter value being the most compatible with the parallaxes and mean distances of the rest of candidate members (Table 1). However, the PMs of Czernik 2 do not match those of its putative companions and no RV has been found, making the membership of this OC to group R highly unlikely.

### 3.2.3. LP 1377/LP 1970

Although LP 1377 has been described by two teams of astronomers using Gaia data [22,27], we did not find such an OC in Gaia EDR3. Instead, we have only found a small eccentric group of a dozen perhaps interrelated stars (Table 1). The original description of LP 1377 included a very high portion of space, considering the low mean parallax. In addition, the $1\sigma$ uncertainties in the PM were unusually large (~0.5 mas/y). However, even if LP 1377 exists, the incompatible RVs of both OCs (Table 1) indicate that they would not be physically related.

### 3.2.4. NGC 2354/NGC 2362

This pair of well-studied OCs would be a good candidate for a binary system from the Gaia dataset (Table 1), although their different ages clearly show that they cannot be a primordial pair. As for the astrometric data, the corrected parallaxes of NGC 2354 converge to a distance of 1.29 kpc, which is within the range of photometric *d*: from 1.26 kpc [26] to 1.39 kpc [44]. Regarding NGC 2362, the parallaxes point to the same distance, which is again within the *d* range: from 1.23 kpc [27] to 1.34 kpc [20]. The projected distance between the two OCs is ~32 pc. Only $\mu_\delta$ show a noteworthy difference. However, the difference in PMs allows us to derive a relative tangential velocity of ~8 km/s, within the adopted limit of 10 km/s. Note, however, that other authors have adopted a limit of 5 km/s (e.g., [8]). Whatever the case, a diagram similar to the one in Figure 1 did not show any correlation between the two OCs. Moreover, the orbital parameters derived by Tarricq et al. [5] do not match. For example, the maximum altitude of the orbit away from the Galactic plane is $175.3 \pm 13.5$ pc for NGC 2354 and $96.5 \pm 13.9$ pc for NGC 2362. Therefore, this questionable binary system candidate requires further study.

### 3.2.5. NGC 581/NGC 663/COIN Gaia 4

This trio of candidates is included in the analysis since Liu and Pang [22] estimated the age of NGC 663 to be 1.1 Gyr. However, a literature search reveals that it is indeed a young OC, with many estimated ages around 0.03 Gyr [20,24,26,27,35,41,45–47]. NGC 663 and COIN-Gaia 4 are also young OCs (Table 1), so even if this triplet was physical, it would be a likely primordial group. However, not all parameters of these clusters are in such good agreement. For example, both the parallaxes and the photometric *d*, which are consistent for each of the OCs separately, indicate that NGC 663 is much further away than COIN-Gaia 4 (Table 1). The Gaia-derived *d* of NGC 663 range between 2.4 kpc [26] and 2.9 kpc [20], while the COIN-Gaia 4 *d* range between 1.9 kpc [26] and 2.2 kpc [20]. The Gaia-derived *d* of NGC 581 are in the range of 2.3 kpc [26] and 2.5 kpc [20], so they are compatible with those of NGC 663, although the latter seems to be further away than NGC 581. The same trend is observed in the parallaxes.

The RV of COIN-Gaia 4 is totally discordant with its supposed companions (Table 1). Therefore, this system cannot be triple. Reported RVs of NGC 581 range from −40.4 km/s [42] to −45.7 km/s [21], and are only marginally compatible with the reported RVs of NGC 663, which range from −31.0 km/s [42] to −33.1 km/s [46]. However, one member star of NGC 663 has a RV of $−44.4 \pm 3.4$ km/s and one star of NGC 581 has a RV of $−44.4 \pm 0.3$ km/s, both from Gaia DR3. Therefore, both NGC clusters form a plausible primordial pair candidate. Incidentally, NGC 581 seems to be part of UBC 186 [20].

On the other hand, it has been proposed that NGC 663 forms a pair with NGC 659 [48], which is also a young OC. However, the astrometric parameters do not match properly (Table 1). The widely varying RVs of NGC 659, from −64.8 km/s [5] to 77.8 km/s [28], are not helpful. Most of the photometric *d* of NGC 659 are between 3.1 kpc [5] and 3.3 kpc [20], indicating that NGC 659 is considerably further away than NGC 663. Correspondingly, all reported parallaxes of NGC 659, once corrected, point to a consistent distance of 3.2 kpc, while the parallaxes of NGC 663 lead to a reliable, shorter distance of 2.9 kpc. Therefore, the suggested link between the two OCs is, at least, doubtful.

### 3.3. Candidate Pairs/Groups Suggested by Piecka and Paunzen

These authors [49] listed a series of 60 OC aggregates from the Cantat-Gaudin and Anders [50] catalogue which share several of the assigned member stars in at least two OCs of the aggregates. About 70% of these candidate pairs/groups are less than 0.1 Gyr old, although aggregates > 0.8 Gyr (from the age of their youngest OC) are also included. We have reviewed that list and analyzed the aggregates with at least one OC of more than 0.1 Gyr. Several of these aggregates could be safely discarded, as their distances and/or PMs are too different to belong to the same physical system. However, some other cases require a deeper analysis.

### 3.3.1. Ruprecht 100/101

Angelo et al. [48] considered this pair a genuine binary cluster. Ruprecht 101 is likely older than Ruprecht 100. The Gaia-derived ages of Ruprecht 101 range from 204 Myr [20] to 559 Myr [26], while those of Ruprecht 100 range from 1.02 Gyr [5] to 1.20 Gyr [22]. However, most parameters of this candidate pair, either from the Gaia literature or from our study, appear compatible at first glance (Table 1). The wide range of *d* for Ruprecht 100, from 2.78 kpc [26] to 3.37 kpc [20], overlaps those for Ruprecht 101: from 2.71 kpc [5] to 2.85 kpc [26]. The same is true for the parallaxes: those of Ruprecht 100, including our measurement, suggest a distance between 3.1 kpc [27] and 3.6 kpc [25], while those of Ruprecht 101 point to a distance between 2.9 kpc [27] and 3.5 kpc [25].

We found two members of Ruprecht 100 with RVs of −11.3 and −11.6 km/s in Gaia DR3. The RV of Ruprecht 101 is very poorly constrained in the literature, ranging from −63.0 km/s [38] to 11.1 km/s [27]. Fortunately, we found 15 members of this OC with Gaia DR3, having RVs from −15.5 to −19.3 km/s, which are compatible with the RVs of Ruprecht 100. On the other hand, the PMs appear too different for such a distant binary cluster. The relative tangential velocity estimated from the PMs and the mean distance is about 18 km/s, well above the acceptable limit for an interacting pair. Moreover, the orbital elements calculated by Tarricq et al. [5] are disparate. Such discrepancies, taken together, seem sufficient to rule it out as a binary cluster.

### 3.3.2. Theia 847/UPK 312

Theia 847 (AKA Alesi 2) is a well-studied cluster, with reported *d* ranging from 501 pc (e.g., [42]) to 657 pc [20]. We derive a photometric *d* of 0.64 kpc (Figure 3), and the set of five Gaia-derived *d* has the same median value. The Gaia mean parallaxes vary between 0.308 mas ([27]; probably erroneous) and 1.62 mas [25]. We also obtained 1.62 mas from Gaia EDR3. Four of these five parallaxes (once corrected for the Gaia DR2 offset) point to a parallax distance of 0.62 kpc. Therefore, there is a good agreement in both independent distances for Theia 847.

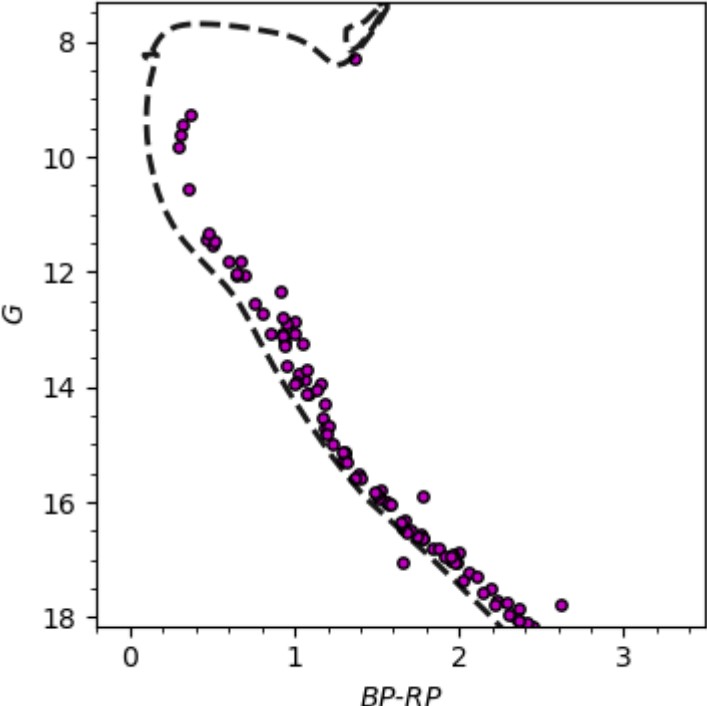

**Figure 3.** CMD and isochrone fitting of our list of probable members of Theia 847, obtained using Gaia EDR3. The constraints are defined in Table 1.

UPK 312 has been less studied. We found only two photometric *d* derived from Gaia: 720 pc [20] and 681 pc [26]. Our own estimate (684 pc) turns out to be the median of the three distances. All reported Gaia parallaxes (once corrected) coincide at 1.42 mas, which is also our median value and corresponds to 0.70 kpc. Again, both distances agree.

If we take the median photometric *d*, UPK 312 is about 44 pc farther away. From the parallax distances, UPK 312 is about 80 pc farther away, so UPK is most likely farther away than Theia 847. If we take the angular distance between the centroids of the two OCs (2.28°) at the nearest cluster distance (0.62 kpc), we get a projected distance of about 25 pc, which combined with the above heliocentric distances, leads to a total distance between them of 50 to 84 pc. This value is too high for a binary cluster (usually less than 30 pc), but it is within our limit for grouped clusters.

Considering now the PM, the difference according to our data is 1.1 mas/y, which corresponds, at the distance of the nearest OC, to a tangential velocity of 3.2 km/s, within the limits adopted for an interacting system. The RVs from Gaia DR3 of nine likely members of UPK 312 range from −8.7 to 4.8 km/s, which are marginally compatible with those of Theia 847: from −13.8 to −7.1 km/s (32 members).

The ages of the two clusters are close enough that they could be born in the same primordial group (Table 1 and Figure 3), but considering the age of the younger OC, the two OCs should already be separated or merged according to theoretical models (e.g., [51,52]). The reported Fe/H of Theia 847 is close to that of the Sun: between −0.045 ± 0.071 [37] and 0.057 ± 0.106 [26], and compatible, within observational error, with the Fe/H of UPK 312: 0.160 ± 0.197 [26].

Therefore, this candidate pair is considered doubtful and further studies, especially on RVs and abundances, are required to confirm or rule it out as an exceptionally long-lived primordial pair.

### 3.3.3. FSR 686/UBC 55 and Gulliver 56/UBC 73

All astrometric and photometric parameters of FSR 686 and UBC 55 are in agreement [20], as we have manually confirmed using Gaia EDR3, except for the number of member stars (107 in our study). Moreover, 11 out of the 12 members of FSR 686 identified by Cantat-Gaudin et al. [20] also belong to UBC 55. In accordance with this, Piecka and Paunzen [49] considered difficult to prove whether this aggregate truly consists of two different OCs. Therefore, it is likely that both objects are part of the same OC.

A similar case was found for the Gulliver 56/UBC 73 candidate pair, since 15 of the 18 members of the first OC are also members of the second [20], and their astrophysical parameters are virtually identical.

### 3.3.4. LP 2117/UBC 354

LP 2117 is a well-populated OC whose reported mean parameters have been confirmed in the present study (Table 1). Although the photometric *d* and *plx* of UBC 354 from Cantat-Gaudin et al. [20] are somewhat different, the remaining parallaxes, after correction for the Gaia DR2 offset, indicate a common distance of 1.61 ± 0.04 kpc for both objects. The RVs of LP 2117 range from −7.6 km/s [5] to −10.0 km/s [27], and the RVs of UBC 354 range from −9.1 km/s [26] to −10.1 km/s [27], i.e., the RVs of both objects are almost coincident. In addition, most of the 22 members of UBC 354 are also members of LP 2117 according to the data of Cantat-Gaudin et al. [20] and the present study. Given their close position and the almost perfect agreement of the rest of the astrometric parameters, we suggest that UBC 354 is part of LP 2117.

Something similar could be happening with some clusters recently detected from Gaia data using unsupervised algorithms in the same line of sight, namely LP 2118, UBC 111, and UBC 574. All of them share compatible positions, parallaxes, PMs, and RVs. According to the data of Hao et al. [27], UBC 111 would be the result of adding LP 2117 and LP 2118. Another clue comes from the contradictory number of UBC 574 members: 9 stars [27]

versus 90 stars [26]. Perhaps these are the reasons why UBC 574 has been removed from the original list of UBC clusters.

### 3.3.5. ASCC 88/Gulliver 29

The Gaia astrometric measurements for Gulliver 29 have been confirmed in our study (Table 1). All corrected parallaxes (including ours) agree on a distance of $1.09 \pm 0.02$ kpc, which is similar to the reported photometric $d$: from 1.119 kpc [53] to 1.18 kpc [20]. The corresponding estimates for ASCC 88 lead to a parallax distance of $894 \pm 10$ pc, which is within the range of Gaia photometric distances: from 864 pc [26] to 931 pc [20]. Even disregarding the projected distance between them (at least 6 pc), we infer that the two clusters are most likely >100 pc apart. Moreover, the $\mu_\delta$ and RV measurements are disparate. The reported RVs of Gulliver 29 range from 1.2 km/s [5] to 29.2 km/s [27], while for ASCC 88, the RV would range from $-38.0$ [5] to 2.8 km/s [24]. All in all, these OCs seem not to be an interacting pair.

### 3.3.6. Barkhatova 1/Gulliver 30

Both clusters appear to be older than 0.1 Gyr (Table 1), although ages as young as 14 Myr [26] and 11 Myr [27] have recently been reported for Barkhatova 1 and Gulliver 30, respectively. Although the RVs of both OCs are marginally compatible, they are poorly constrained. For Barkhatova 1, we find RVs from $-37.3$ km/s [27] to $-1.0$ km/s [25], and for Gulliver 30, we have $-10.2$ km/s [8] and 13.9 km/s [5].

The rest of the astrophysical parameters seem compatible at first glance, but a closer examination reveals some inconsistencies in the parallax and $\mu_\delta$. The Gaia photometric $d$ for Barkhatova 1 range from 1.86 kpc [26] to 2.19 kpc [20], and for Gulliver 30, they range from 2.03 kpc [26] to 2.27 kpc [20]. Therefore, the $d$ is similar but appears larger for Gulliver 30 on average. On the other hand, the mean parallaxes are significantly different and also show that Gulliver 30 is further away. Hunt and Reffert [54] found disjoint parallax ranges for Barkhatova 1 (0.456–0.469 mas) and Gulliver 30 (0.421–0.451 mas). Six out of seven Gaia parallaxes for Barkhatova 1, after correction of DR2 values, lead to a consistent distance of $2.02 \pm 0.04$ kpc, within the range of photometric $d$. Similarly, for Gulliver 30, we have an average distance of $2.21 \pm 0.05$ kpc, also within the photometric range. Considering a projected distance of at least 7 pc, the total distance between the two OCs is probably $\geq 100$ pc, so it is most unlikely that these OCs form a physical pair.

### 3.3.7. NGC 2168/Coin-Gaia 24

This pair includes the famous OC M35 (NGC 2168). Despite the many studies on M35, its age is poorly constrained. Ages ranging from 0.06 Gyr [55] to 0.40 Gyr [40] have been published. If we restrict ourselves to Gaia-derived ages, the records range from 0.09 Gyr [22] to 0.40 Gyr [40]. The four Gaia ages of Coin-Gaia 24 are scattered in a similar range: from 0.06 Gyr [22] to 0.25 Gyr [26]. Therefore, this pair could be (or not) young by our criteria. The PMs are also similar (Table 1), but the only reported RV of Coin-Gaia 24 (23 km/s; [38]) is at odds with the RV of NGC 2168.

Fortunately, distances and parallaxes help to clarify this case. Two photometric $d$ of Coin-Gaia 24 have been found: 1.03 kpc [20] and 0.94 kpc [26]. All reported parallaxes—from 0.956 mas [22] to 0.989 mas [27]—lead, after offset correction, to a consistent distance of $1.011 \pm 0.005$ kpc. For M35, we found 11 photometric distances ranging from 0.80 kpc [56] to 0.91 kpc [20,40]. The median of these distances is 0.85 kpc [57]. All reported parallaxes—from 1.123 mas [22] to 1.156 mas [27]—point to a consistent distance of $0.864 \pm 0.004$ kpc. Consequently, it is safe to conclude that M35 is more than 100 pc closer to the Sun than Coin-Gaia 24. Therefore, the two OCs do not form an interacting pair, although some of their characteristics suggest that they may have been born in the same complex.

### 3.3.8. Kronberger 1/Gulliver 53/Stock 8/NGC 1893/Coin-Gaia 40 (Group V)

Although the astrometric parameters of Stock 8 and Gulliver 53 look similar at first glance, there are some inconsistencies in the distances and PMs. The Gaia photometric $d$ for Stock 8 ranges from 2.03 kpc [26] to 2.35 kpc [20]. The Gaia corrected parallaxes agree on a distance of 2.12 ± 0.02 kpc, within the $d$ range. On the other hand, the $d$ reported for Gulliver 53 ranges from 2.19 kpc [26] to 2.43 kpc [20], and all corrected parallaxes (including our measurement) are consistent with a distance of 2.48 ± 0.06 kpc. The RV of Stock 8 and Gulliver 53 are very poorly constrained: from −18 km/s [24] to 42 km/s [5] and from −35.6 km/s [27] to 30 km/s [37], respectively. Nonetheless, this dataset suggests that Gulliver 53 is too far apart from Stock 8 to form an interacting pair.

The only age found for Coin-Gaia 40 is 0.14 Gyr [26], and just one estimated photometric distance has been found: 1.92 kpc [26], which is compatible with the distance derived from five corrected parallaxes: 2.0 kpc. Therefore, this OC may match Stock 8 but not Gulliver 53. On the other hand, the PMs of Coin-Gaia 40 match with Gulliver 53 but not with Stock 8. No reliable RV has been found for Coin-Gaia 40. Therefore, its membership in group V is doubtful. Moreover, there is no evidence for the membership of NGC 1893 in the group.

On the other hand, the membership of Kronberger 1 (=Alicante 12) and NGC 1931 to the Stock 8 group is probable by any standards (Table 1), even if the RV of Kronberger 1 is poorly known, ranging from −21.2 km/s [37] to 13 km/s [24]. NGC 1931 is a very young cluster, with reported ages ranging from 2 Myr [24] to 12 Myr [58]. It has not been well studied, as it is still embedded in its parent nebulosity, leading to a too wide MS in the CMD and to unreliable photometric $d$. However, we obtained some approximate astrometric parameters from Gaia EDR 3 (Table 1) and a RV of −22.5 ± 15.1 km/s for one source (#182583805697358464) from Gaia DR3, which agree with those of Stock 8 and Kronberger 1. Incidentally, NGC 1931 is the same OC as FSR 791, although the latter is catalogued as a different object at the Simbad database. Note that all these group members are very young, so we have a characteristic primordial group.

### 3.3.9. Collinder 220/IC 2581

Collinder 220 is significantly older than IC 2581. The astrometric parameters are similar, with differences in $plx$, $d$, and $\mu_\delta$ (Table 1). Both parallaxes and $d$ suggest IC 2581 is further away than Collinder 220. For IC 2581, all corrected parallax distances are 2.66 ± 0.04 kpc. Our determination of the median parallax of Gaia EDR3 from the 43 most accurately measured stars (0.374 mas) leads to a coincident distance of 2.67 kpc. These values are slightly larger than $d$ in Table 1, but a Gaia $d$ of 2.62 kpc has also been published [8]. For Collinder 220, the parallax distance is 2.56 ± 0.07 kpc. We obtained a median parallax (43 best quality members) of 0.392, which points to a concurrent distance of 2.55 kpc. Again, these values are somewhat higher than $d$ in Table 1, but a $d$ of 2.75 kpc has also been reported [53]. However, the differences found cannot rule out the possibility that the distance between the two objects is less than 100 pc, even considering a projected distance of ~17 pc. On the other hand, we have a difference in PMs of 0.92 mas/yr that, at the average distance of the system, would imply a relative tangential velocity > 11 km/s, too high for a physically interacting system. The different RVs also suggest the same conclusion, although for IC 2581 an RV as low as −4.6 km/s has been repeatedly quoted [21,31,35,46]. Even if this minimum value proved correct, the difference in RVs would also push the difference in absolute velocities to ~14 km/s, making this pair an unlikely physical system. Accordingly, the orbital elements of both OCs do not overlap [5].

### 3.3.10. FSR 448/FSR 451

Our reassessment of FSR 451 agrees with the reported mean astrometric parameters within the observational uncertainties. Although both OCs have very different ages, they could be close enough in space, according to their $d$ and $plx$ (Table 1). However, the PMs and RVs are significantly different. PMs and parallaxes allow inferring a difference in

tangential velocity of at least 11 km/s. The difference in RV is even larger, making this pair rather implausible as a binary system.

### 3.3.11. NGC 2345/FSR 1207

Most of the relevant parameters of NGC 2345 and FSR 1207 agree well, with the possible exception of $\mu_\delta$ and RV. We re-examined FSR 1207 using Gaia EDR3 and found impeccable agreement with the published data (Table 1). Although the mean parallax appears somewhat larger, our median value (0.380 mas) is in perfect agreement with the parallax of Poggio et al. [25]. This value also happens to be the median of the reported parallaxes. They all agree and point to a distance of 2.63 ± 0.02 kpc. The reported photometric $d$ is 2.51 kpc [20] and our $d$ is 2.58 kpc. Therefore, the three values are in reasonable agreement. The corrected Gaia DR2 parallaxes of NGC 2345 converge to a distance of 2.66 kpc, while the EDR3 parallaxes (including our median $plx$: 0.357 mas) suggest a distance of 2.80 kpc. Its Gaia photometric $d$ ranges from 2.43 kpc [26] to 2.74 [44] kpc. These estimates suggest that NGC 2345 is slightly further away than FSR 1207. However, the distances of both OCs could be compatible.

From the differences in PM and an assumed pair distance of 2.6 kpc, we estimate a relative tangential velocity of 8 km/s, which is within our adopted criteria. Unfortunately, Gaia DR3 has not released any RVs attributable to probable FSR 1207 members. The only reported RV of FSR 1207, 29.4 km/s [44], would rule out this pair if confirmed, as RVs of NGC 2345 range from 53.8 km/s [44] to 63.3 [5], and we have found four members of NGC 2345 with RVs of 58–59 km/s. Otherwise, the comparable ages of both OCs would allow this to be an exceptionally long-lived primordial pair. Unfortunately, the metallicity of these OCs does not help in this case, as the metallicity of FSR 1207 has not been reported, to our knowledge. Finally, let us point out duplicity in the Simbad database: NGC 2345 and FSR 1206 are recorded as different OCs, but are the same object.

### 3.3.12. Pismis 19/Trumpler 22/NGC 5617/Hogg 17 (Group X)

De Silva et al. [59] performed a photometric and spectroscopic study of the OCs NGC 5617 and Trumpler 22 to conclude that they have an age of 0.07 Gyr and form a binary cluster. Bisht et al. [60] reached the same conclusion, identified the member stars from Gaia data, and estimated the Galactic orbits for both OCs, inferring ages of 0.09 Gyr. Angelo et al. [48] concluded that NGC 5617 and Trumpler 22 have the same distance, age, and compatible metallicities, indicating they are a physically interacting pair with a common origin.

A possible link between Pismis 19 and Trumpler 22, suggested by de la Fuente Marcos and de la Fuente Marcos [6], has already been ruled out [9]. Pismis 19 is probably a background cluster with a markedly higher interstellar reddening than the rest of the group [48].

Gaia-based studies on Trumpler 22, including our work, converge on a photometric $d$ between 2.2 kpc [26] and 2.4 kpc [5,8,20]. Such $d$ range agrees with the measured parallax, taking into account the global offset of the Gaia DR2 parallaxes [30]. All reported parallaxes, including our measurement, lead to distances from 2.3 kpc [27] to 2.6 kpc [60]. NGC 5617 has $d$ ranging from 2.0 kpc [26] to 2.5 kpc [60]. All obtained parallaxes point to a distance of 2.36 ± 0.07 kpc. For Hogg 17, we find $d$ from 2.2 kpc [5,26] to 2.3 kpc [8]. Reported parallaxes indicate distances from 2.25 kpc [27] to 2.40 kpc [25]. Whatever the case, we note that the dataset for these three OCs is compatible with a mean distance to the group of 2.3 kpc.

The PMs are very similar for Trumpler 22 and Hogg 17 (Table 1). However, the differential PMs between Hogg 17 and NGC 5617 suggest a tangential velocity close to the 10 km/s limit. The RVs of NGC 5617 range from −34.7 km/s [8] to −38.6 km/s [24]. From the difference with the RV of Hogg 17 (Table 1), we determine that both OCs move apart by at least 11 km/s, which casts doubt on the membership of NGC 5617. However, even if NGC 5617 is part of the group, the Primordial Group hypothesis would not be necessarily falsified, since most of the reported ages of NGC 5617 are ≤ 0.10 Gyr.

On the other hand, Trumpler 22 and Hogg 17 seem to be well-matched in all categories. The RVs of Trumpler 22 range from −38.5 km/s [24] to −43.1 km/s [5,26]. In addition, we have found three Trumpler 22 members that have RVs practically equal to those reported for Hogg 17, and the small differences in the PMs lead to a tangential velocity < 3 km/s. Regarding the age, both OCs could be similar and not as old. The age of Trumpler 22 is poorly constrained. Pre-Gaia studies report ages down to 0.31 Gyr [35]. However, the catalog of Kharchenko et al. [35] is not very suitable as a source of ages for young OCs, as the relevant parameters are based on near-infrared photometry (2MASS), and the corresponding CMDs have low age sensitivity in this age interval. Anyway, the most recent works report ages close to 0.03 Gyr [5,20,40]. The age of Hogg 17 is also not well known. Reported values range from 0.07 Gyr [57] to 0.18 Gyr [5]. Nevertheless, the most usual ages are ca. 0.10 Gyr (e.g., [24,26,31,41,61];). The CMDs of both OCs are well-matched [20], suggesting that they may form a primordial pair of similar age and composition.

### 3.3.13. NGC 2194/Skiff J0614 + 12.9

This candidate pair seems to have compatible parameters (Table 1). However, the differences in PMs at such a long distance imply a relative tangential velocity of ~11 km/s. As for the RV of Skiff J0614 + 12.9, Gaia studies agree on a value of 45.6 ± 0.3 km/s [5,26,27]. Remarkably, the same authors find different RVs for NGC 2194: from 12.4 km/s [27] to 41.6 [26]. The variation may be related to the diverse lists of member stars. While Hao et al. [27] report only 169 members, Dias et al. [26] include 795 probable members. In any case, we found five stars of NGC 2194 with RV, all in the range of 9.0 km/s to 15.4 km/s, in good agreement with several works [20,24,27,31,35,38,56]. Even if the higher RV value is correct and despite the compatible distances and ages, it is safe to say that the two OCs are likely not a binary system, as their differential velocity is well above the undemanding limit of 10 km/s.

### 3.3.14. Ruprecht 44/43 (Group Y)

This candidate pair has recently been studied without reaching a definitive conclusion [3]. Both OCs are at the limiting distance to obtain accurate parallaxes and PMs from Gaia data. Consequently, the reported parallaxes for Ruprecht 44 range from 0.152 mas [20] to 0.255 mas [22]. The offset-corrected parallaxes for Gaia lead to a median distance of 5.4 kpc [26], compatible with the range of photometric *d*: from 4.6 kpc [31] to 5.8 kpc [20]. There is a broad consensus on its young age: ~0.01 Gyr [20,24,55,57]. The only retrieved RV of Ruprecht 44 was 94 ± 30 km/s, from a single star [24]. We obtained a mean RV of 71 ± 10 km/s from published data on 12 OC B-type stars [3,62]. Three candidate member stars [50] also point to a mean RV of 71 ± 26 km/s. A possible member from our study, but 9 arcmin away from the cluster center, has an RV of 77 ± 6 km/s.

Ruprecht 43 is an intriguing object. For example, Cantat-Gaudin et al. [20] have only reported one member star with probability ≥ 0.7, and the reported parallaxes vary widely from 0.105 mas [20] to 0.381 mas [27]. Considering only the literature data, it is not sure whether or not Ruprecht 43 is related to Ruprecht 44. The Gaia astrometry of both OCs coincides (Table 1). The RV of Ruprecht 43 would range between 104 km/s [5,8] and 113 km/s [24,35]. Only the Kharchenko team has reported the age of Ruprecht 43 to be ~0.25 Gyr (e.g., [35]). If correct, the age disparity would falsify our hypothesis in case the two OCs are related (see, however, the warning about the ages in Kharchenko's catalog in the subsection on Group X). Generally, we should be cautious with pre-Gaia estimates of OC parameters [24]. For example, distances of Ruprecht 43 as low as 970 pc were reported not long ago [63], in conflict with the mean parallax obtained from Gaia EDR3.

To address this case, we obtained a list of 36 probable members of Ruprecht 43 with a median parallax of 0.18 mas (within the published values; the parallax values in Table 1 only reflect the range of member parallaxes). We derive a photometric *d* of 5.5 kpc, fully compatible with this median parallax and the quoted *d* of Ruprecht 44. Assuming such a distance to the system, the projected distance between the two OCs would be only ~24 pc.

Our estimate of the Ruprecht 43 extinction is $A_V$ = 1.3 mag. One of its member stars (#5597405481305457920) has an RV in Gaia DR3 of 94.6 ± 0.3 km/s, somewhat lower than previous literature, but compatible with RVs reported for Ruprecht 44. Moreover, the CMDs of both OCs fit sensibly well, and our age estimate, 0.05 Gyr, suggests that this would be a young OC that could belong to the primordial group of Ruprecht 44 (Figure 1). There is also a minor (r < 1.2 arcmin) group of 15 stars at Galactic coordinates 246.12, 0.36, i.e., 11 arcmin from the center of Ruprecht 43 [55]. Based on their close astrometry, it could be another associated member and/or a second disaggregated core of Ruprecht 43. However, this preliminary conclusion needs confirmation.

### 4. Conclusions

The Primordial Group hypothesis postulates that only sufficiently young OCs can be multiple, and old OCs are essentially single since the gravitational interaction between OCs in primordial groups is very weak and the probability of gravitational capture of two OCs without disruption or merger is very low. We test that hypothesis through manual mining of Gaia EDR3 and careful revision of the extensive literature on OCs.

To the best of our knowledge, we have revisited all the double/multiple clusters candidates in the literature having any member older than 0.1 Gyr and found no convincing evidence of any old binary system in the Galaxy (see also Casado [9] and Anders et al. [16]).

In the present paper, we complete the test of said hypothesis by revising a total of 120 pairs or groups of clusters and obtaining the same main result. Most of the systems are optical pairs or flyby encounters. However, we also found three dubious pairs (P, R, and S in Table 1) that could falsify the title hypothesis upon further research. Thus, we consider them especially interesting to follow up on. Furthermore, we found two possible primordial pairs older than expected from theoretical models (pairs U, 0.3 Gyr, and W, 0.2 Gyr). We found five other primordial groups, most of them having flawed or uncertain ages but most likely within the prescribed maximum age ~0.1 Gyr (Q, T, V, X, and Y in Table 1). In this respect, we have to warn about the frequent overestimations of young OCs ages in the Kharchenko et al. [35] catalogue. Two of the reviewed objects (LP 1377 and Loden 195) are probably not physical OCs. Last but not least, three of the candidate pairs (FSR 686/UBC 55, Gulliver 56/UBC 73, and LP 2117/UBC 354) are most probably single OCs, in line with a similar conclusion in the first part of the present series [9].

Unexpectedly, we also detected that numerous FSR OCs, classified as new at CDS databases, are duplicities since they correspond to well-known OCs. Some examples are FSR 455 (=Berkeley 99), FSR 457 (=King 12), FSR 458 (=NGC 7788), FSR 461 (=NGC 7790), FSR 467 (=King 11), FSR 468 (=NGC 7762), FSR 475 (=King 13), FSR 490 (=King 1), FSR 491 (=NGC 103), FSR 694 (=NGC 1605), FSR 791 (=NGC 1931), FSR 1206 (=NGC 2345), and FSR 1287 (=NGC 2362). Other detected duplicities are UBC 175 (=NGC 7438) and UBC 284 (=Patchick 57).

Our results confirm that the vast majority of binary/multiple OCs in the Galaxy, if not all, are of primordial origin and are disaggregated within less than a Galactic rotation, in line with observational and theoretical studies of the Magellanic Clouds [7,13,14,52] and theoretical *N*-body simulations in the galaxy [11,51,64]. The pairs of OCs in these groups are generally not binary systems since they are not gravitationally bound. The Primordial Group hypothesis has successfully passed all the tests so far.

**Funding:** This research received no external funding.

**Informed Consent Statement:** Not applicable.

**Data Availability Statement:** The data associated with this manuscript are available in the public data repositories mentioned in the Acknowledgements section and in the referenced studies.

**Acknowledgments:** Thanks to Alfred Castro Ginard for providing updated astrophysical parameters and figures of some of the OCs studied. This work made use of data from the European Space Agency (ESA) mission Gaia (https://www.cosmos.esa.int/Gaia, accessed on 7 October 2021), processed by the Gaia Data Processing and Analysis Consortium (DPAC, https://www.cosmos.esa.int/web/Gaia/dpac/consortium, accessed on 7 October 2021). Funding for the DPAC was provided by national institutions, in particular the institutions participating in the Gaia Multilateral Agreement. This research made extensive use of the SIMBAD database and the VizieR catalogue access tool, operating at the CDS, Strasbourg, France (DOI: 10.26093/cds/vizier), and of NASA Astrophysics Data System Bibliographic Services.

**Conflicts of Interest:** The author declares no conflict of interest.

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
