# Peer review of "The Effect of Age on the Grouping of Open Clusters: II—Are There Old Binary Clusters?"

_universe, doi:10.3390/universe8070368_

Round 1
Reviewer 1 Report
This is the report on: "The Effect of Age on the Grouping of Open
Clusters: II. Are there old binary clusters" by Juan Casado.
The manuscript discusses the possible binarity of a number of nearby
open clusters. Some of these clusters appear optically paired, but it
is not known if they form a bound pair. The author analyses Gaia EDR3
data and establishes three real pairs and possible a few other
potential candidates among 22 groupings.
In the manuscript, the author discusses each pair of clusters and
review their potential for boundness, or the lack thereof.
The manuscript is well written and to the point. The results are
interesting, although I think that the analysis and conclusions could
improve by performing a k-nearest neighbor analysis technique in order
to establish the association among the star clusters. The author
discusses a few theoretically and simulation-oriented publications on
cluster pairs, and for completeness, it would be good if a reference to
the original paper by Portegies Zwart and Rusli (2007MNRAS.374..931P,
on isolated binary clusters). And, also it discusses LMC paired
clusters, a reference to Mucciarelli et al (2012ApJ...746L..19M) would
strengthen the arguments in the manuscript.
Reviewer 2 Report
This paper completes the investigations on open cluster groups
started with paper I (Casado 2022, Universe 8, 113) by the same author.
The author aims at confirming his hypothesis, postulated in paper I,
that there are no binary or multiple open clusters older than about
100 million years. After first tests already carried out in paper I,
the author checked all remaining open cluster groups known from the
literature. This was done carefully and described in detail, going
through the list of candidates one by one. I have no major concerns
about this paper, which is generally well written, but only a few
minor comments and questions, which the author should take into
acount, before this paper can be accepted for publication in Universe.
Minor points:
Is the number of ''120 pairs/groups revised'' the number investigated
in this paper or the total number studied in both paper I and this paper?
There are only 23 pairs/groups incl. 56 individual clusters listed in
Table 1 and a few more (4 pairs) discussed in Section 3, but not
included in Table 1. Does that mean that out of 120 pairs/groups,
discussed in the literature, only 27 had an overlap in their data as
desribed in the 2nd paragraph of Sect.2 ?
It would be nice to give some statistics on the obviously large number
of literature pairs/groups of clusters not even discussed in this paper
and mention the main reason (probably the very uncertain pre-Gaia
proper motions) for excluding them already during or after the first
steps of the methodology (check for projected distance of less than
100 pc, and comparison of astrometric data within 3 sigma).
In the last paragraph of Section 1, the sentence ''To do this, we have
used the Gaia EDR3 data and the numerous references available.'' is not
clear to me. Are these references also based on EDR3? All references
in column 13 of Table 1 apparently used EDR3. The note $^e$ in Table 1
is used for ''This work'' only, but may be it is true for all entries?
It would be easier for the reader, if all cluster groups were sorted
in Table 1 by the same order as they appear in Section 3. In one case,
NGC 2168 and Coin-Gaia 24, the first column in Table 1 is empty (should
be ''-''?).
In Section 2, the discarded sources have $G > 18$,
whereas in Fig. 3 the selected sources have $G_{mag} < 19$.
Concerning Section 2, paragraph starting with ''Following the criteria of
previous studies ...'', was the 100 pc limit in previous studies and in
this study always used for the projected distances only, or were the new
distances between the members of different clusters computed in 3D space
based on Gaia?
Figure 3 would, in my opinion, be more useful with additional features:
1) showing the limit for projected separation of 100 pc, e.g. with a bar,
2) drawing circles (ellipses) of the cluster radii in arcmin
The first cluster groups discussed in Section 3 may also go to a
first subsection (''Candidate groups/pairs suggested before Gaia'' ?)
before the following two subsections.
Reviewer 3 Report
As the 2nd paper of the same series submitted to this journal, the author further tests the Primordial Group hypothesis, which was defined in Casado (2022).
This is an important work as it directly addresses the formation and evolution of open clusters in the Milky Way. It is also a difficult task as the analysis includes a huge number of open clusters. The author has done an excellent job and confirmed that the majority of binary/multiple open clusters in the Galaxy are of primordial origin and are not stable for a long time, in line with the results in theoretical N-body simulations. Therefore, the current paper validates the Primordial Group hypothesis.
I strongly recommend that this paper can be accepted in present form.
